# A Systematic Review of LET-Guided Treatment Plan Optimisation in Proton Therapy: Identifying the Current State and Future Needs

**DOI:** 10.3390/cancers15174268

**Published:** 2023-08-25

**Authors:** Melissa McIntyre, Puthenparampil Wilson, Peter Gorayski, Eva Bezak

**Affiliations:** 1Allied Health & Human Performance Academic Unit, University of South Australia, Adelaide, SA 5000, Australia; 2Department of Radiation Oncology, Royal Adelaide Hospital, Adelaide, SA 5000, Australia; 3UniSA STEM, University of South Australia, Adelaide, SA 5000, Australia; 4Australian Bragg Centre for Proton Therapy and Research, Adelaide, SA 5000, Australia; 5Department of Physics, University of Adelaide, Adelaide, SA 5005, Australia

**Keywords:** proton therapy, linear energy transfer, optimisation, treatment plan

## Abstract

**Simple Summary:**

The increasing global demand and accessibility for proton beam therapy has been fueled by its recognised clinical benefits over traditional radiotherapy, primarily its ability to precisely target tumors while sparing more healthy tissue. However, the effectiveness of proton treatment plans can be compromised by four key variables: patient motion, patient setup, proton range, and radiobiological effects. A promising solution for mitigating radiobiological uncertainty is a novel method known as Linear Energy Transfer (LET) optimisation. This systematic review is aimed at compiling data related to the clinical motivation for, and various methodologies of, LET optimisation in proton beam therapy. It will present the current standing of LET optimisation in the field, and provide recommendations for future research and clinical deployment.

**Abstract:**

The well-known clinical benefits of proton therapy are achieved through higher target-conformality and normal tissue sparing than conventional radiotherapy. However, there is an increased sensitivity to uncertainties in patient motion/setup, proton range and radiobiological effect. Although recent efforts have mitigated some uncertainties, radiobiological effect remains unresolved due to a lack of clinical data for relevant endpoints. Therefore, RBE optimisations may be currently unsuitable for clinical treatment planning. LET optimisation is a novel method that substitutes RBE with LET, shifting LET hotspots outside critical structures. This review outlines the current status of LET optimisation in proton therapy, highlighting knowledge gaps and possible future research. Following the PRISMA 2020 guidelines, a search of the MEDLINE${}^{®}$ and Scopus databases was performed in July 2023, identifying 70 relevant articles. Generally, LET optimisation methods achieved their treatment objectives; however, clinical benefit is patient-dependent. Inconsistencies in the reported data suggest further testing is required to identify therapeutically favourable methods. We discuss the methods which are suitable for near-future clinical deployment, with fast computation times and compatibility with existing treatment protocols. Although there is some clinical evidence of a correlation between high LET and adverse effects, further developments are needed to inform future patient selection protocols for widespread application of LET optimisation in proton therapy.

## 1. Introduction

The dosimetric advantages of proton beam therapy (PBT) include increased sparing of critical structures, with the potential to increase tumour control, as the delivered radiation dose can be escalated. This is achieved through the ability to modulate and shift the Bragg Peak, leading to high dose conformality to the target and normal tissue sparing. It is hence favourable for challenging cases where the target is in close proximity to critical structures. Its widespread demand and availability have expanded rapidly [1], with improvements to technology and modern delivery techniques increasing its efficacy and affordability.

Our ability to select a beam energy that penetrates and stops in the distal region of the tumour is unique to heavily charged particles, such as protons and heavy ions. Conformality is constantly improving through intensity-modulated PBT (IMPT) using the pencil-beam scanning (PBS) technique, both of which have become clinical commonplace. However, uncertainties such as patient setup and motion as well as proton range and radiobiological effects degrade this conformality. Therefore, research into mitigating these uncertainties is of significant interest. Patient motion, setup and proton range uncertainties are addressed through robust optimisation (RO) approaches to treatment planning [2] as well as other motion-management techniques [3]. However, biological uncertainties still remain largely unaddressed.

Radiobiological uncertainty is potentially a major contributor to PBT uncertainty, arising from the increase in relative biological effectiveness (RBE) in the distal fall-off region of the Bragg Peak. A universal RBE of 1.1 is currently adopted in a clinical setting [4,5], despite increasing to values in excess of 2 [6] in the distal Bragg Peak region. Clinically, the distribution of RBE for a treatment plan is complicated further by the use of the spread-out Bragg Peak (SOBP), due to the higher degree of modulation compared to a pristine Bragg Peak (e.g., IMPT) [7,8,9,10]. By disregarding RBE exceeding 1.1 in the distal region, we underestimate the biological effect inside critical structures, potentially having a negative effect on treatment outcomes [11]. A universal RBE of 1.1, for both dose reporting and treatment planning, has been decided by consensus of clinical specialists and originates from a global average taken across the results of numerous in vitro and in vivo experiments, such as those reviewed by Paganetti et al. [12]. These experiments have shown that RBE is dependent on physical and biological parameters, such as linear energy transfer (LET), or the ionisation rate over the proton track, dose-fractionation, dose-rate and tissue-sensitivity (e.g., α/β) [13,14]. Although variable RBE models have been proposed [15], their predictions are model-dependent and have significant uncertainty [14]. It is clear that a universal scaling factor is an oversimplification of the radiobiological effect of PBT. However, as variable RBE models are currently unable to accurately reflect biological response in a conventionally absorbed dose-based treatment plan [16,17,18], they may not be suitable for clinical implementation.

As an alternative to RBE-based treatment plan optimisation, LET-based optimisation has been proposed [19,20], as high RBE is primarily a consequence of high LET [13]. Thus, LET is viewed as an appropriate first-order approximation of RBE, especially in the clinically relevant LET range [14,21]. This is a useful approximation to make because LET is a directly observable quantity that can be measured experimentally [22], or computed via analytical and Monte Carlo methods [23]. By optimising PBT treatment plans with respect to dose and LET, one can arrive at a biologically optimal plan within an acceptable margin of error. The general intent of LET optimisation is to shift high LET outside organs at risk (OARs) and, where achievable, inside the target.

As there has been significant interest in LET optimisation of PBT recently, this review aims to summarise aspects of LET optimisation with regard to clinical benefit and the various approaches available. Aspects of each method, such as dosimetric and LET changes achieved and clinical compatibility, are compared and discussed. In the process, the current status of LET optimisation in PBT is summarised whilst highlighting any literature gaps and scopes for future research.

## 2. Materials and Methods

This systematic review was conducted in line with the PRISMA 2020 guidelines [24] and was not registered with any registry. The search strategy using the MEDLINE${}^{®}$ and Scopus databases, summarised in Section A.1, was performed in July 2023, following review by an academic librarian. A grey literature search was also performed using Google Scholar, with the first 10 pages of search results extracted for screening. After removing duplicates, conference abstracts, non-English language articles, articles published prior to 2000 and independent screening of the remaining abstracts by two reviewers (M.M. and E.B.), a total of 131 papers were left for consideration of eligibility. In the event of disagreement between the two reviewers, a third opinion was sought (P.W. or via consensus discussion). Figure 1 outlines the article inclusion process in each phase.

The full-text review was performed using the inclusion/exclusion criteria outlined in Section A.2, which was dictated in advance. At this stage, it was decided that henceforth only articles on PBT would be considered, since variable RBE optimisation is already clinically implemented in heavy-ion treatment planning [25,26]. Data extraction was performed on the remaining articles, where patient, treatment planning and optimisation details were tabulated for comparison and synthesis.

The results commence with a discussion of the clinical data and the correlation between LET and adverse effects (AEs). An in-depth discussion of variable RBE correlations with AEs is found in Underwood et al. [27]; however, in the current review, a discussion of only LET correlations is performed with regard to LET optimisation. This includes support for/against clinical implementation, and possible sources of LET-based objectives for specific OARs to be used in future implementations of LET optimisation. This is followed by a discussion of how LET can be optimised through delivery technique or integration of LET-based objectives into the standard dose objective function. Due to the diversity of the metrics reported across the included studies for treatment plan evaluation, a meta-analysis could not be performed. Instead, a discussion of whether the objectives of each optimisation were met, and how effectively, was performed. The various advantages and drawbacks of each method are discussed with respect to clinical implementation and feasibility.

This review does not focus on LET calculation methods; however, note that LET is an average per particle and, as such, is dependent on the averaging method used. The major LET averaging methods are “dose-averaging”, i.e., average distance over which a fixed energy is imparted, or “track-averaging”, i.e., average energy imparted over a fixed length. Dose-averaged LET yields a different LET to the track-averaged variant [9,28] and is also dependent on which particles are included in the computation (e.g., secondary particles, such as neutrons). All studies included in this review use the dose-averaged variant, unless stated otherwise. Further, the impact of including or omitting secondary particles for LET computation has been extensively reviewed by Kalholm et al. [23] and is thus not discussed in this review. Efforts have aimed to harmonise LET definitions across PBT centres [29]; however, the studies included in this review use dose-averaged LET, which will henceforth be referred to as LET. A LET computation is also typically performed on a microscopic scale (keV/μm). In treatment planning, a single CT voxel is often 1–3 mm${}^{3}$ in size and the LET variance within is significant. A review by Deng et al. [19] suggested lineal energy is a more biologically representative quantity to this respect. However, the high computational demand associated with LET calculations is already a challenge in a clinical setting. Therefore in this review, the focus will be the potential clinical benefit and compatibility with a clinical environment, and not the computation of LET itself.

During the full-text review, LET optimisation articles were classified as such only if they optimised over LET directly or via the “LET-weighted dose” or “biological surrogate dose” (BD). This is to account for the inverse relationship between dose and LET to ensure that any LET changes occur on a clinically relevant scale. We compute the total dose as the sum of the physical ($D_{\mathrm{phys}}$) and biological (BD) dose as follows:(1)$$D_{\mathrm{phys}}+BD=D_{\mathrm{phys}}\times \left(1+c\times LET\right)\;\left[\mathrm{Gy}\right]\;,$$
where *c* is a scaling parameter (μm/keV) that sets the middle of a 5 cm modulated SOBP with a 10 cm range to an RBE of 1.1 [18,30,31,32,33,34,35]. McMahon et al. [18] has shown that using the BD metric in place of RBE yields a lower biological variability in treatment planning and is independent of tissue-specific parameters, such as α/β. Therefore, studies that have optimised over BD [30,31,32,33,34,35] in place of LET alone are included in this review.

## 3. Results and Discussion

### 3.1. Clinical Data and Evidence

The primary question to answer with regard to LET optimisation is if there is a clinical benefit and, if so, which patient cohorts would see the most benefit. Currently, clinical trials are in short supply and thus evidence of clinical benefit is limited. Although trials are underway [36,37], deriving meaningful results will take time. The studies considered in this review aim to measure the correlation of LET with specific AEs. Using retrospective analyses of PBT plans delivered to patients, they assess LET-AE correlations by comparison with follow-up images. These studies do not aim to demonstrate the clinical benefit of LET optimisation directly, but rather, to motivate the need for it.

An informative discussion in Underwood et al. [27] highlighted limitations in decoding the LET dependence from other parameters such as clinical data (e.g., age and gender), diversity in the images recorded and how well the AEs are featured, and the effect of small sample sizes. Whilst there is overlap in the following discussions, the focus is the LET-AE correlation and consequences for LET optimisation.

Table 1 summarises the patient cohorts, treatment details, the clinical endpoint investigated, follow-up times and key findings of 20 studies. The type of analysis used is dependent on the nature of the AE under investigation. AEs that are spatially detectable include follow-up magnetic resonance (MR) image changes or lesions [17,38,39,40,41,42,43,44,45], necrosis [46,47], areas of recurrence (i.e., secondary cancer) [48] and bone fractures [49], which are compared with the LET distribution of the delivered plan. Whereas symptomatic effects such as dysphagia [50], brainstem or CNS injury [51,52], blindness [17] and rectal bleeding [53] warrant a temporal LET-volume-based analysis. The conclusions of these studies, however, are contradictory, citing numerous limitations.

A major limitation common amongst all studies in Table 1 is the retrospective nature, where results are based on patients treated as long as 20 years ago. Since then, new delivery techniques have rapidly expanded (e.g., IMPT with PBS) and are known for producing less trivial, higher LET distributions [54]. More modern PBS delivery methods are associated with LET values as high as 12 keV/μm [55], whereas passively scattered (PS) PBT can only reach up to 8 keV/μm [9] due to a higher energy degradation. Of the five studies analysing patients who received PS PBT for brain, head and neck (H&N) and base of skull tumours [39,46,48,52,56], two reported correlations of MR image changes with LET, only one of which was a strong correlation. Currently, the most common delivery technique is PBS, with 11 studies in Table 1 stating it as the technique used. Nine of these studies suggested that the clinical endpoint is associated with an elevated LET or BD in the AE region or volume. At a glance, this suggests that AEs are observed more frequently in PBS patients; however, this is often only one contributor of many to AEs, and retrospective analyses limit our ability to elucidate the other contributing factors.

Historically, understanding the relative contributions of RT treatment parameters to AEs is a primary focus of research. LET-optimised planning is still a novel technique, and as such, the benefits are widely unknown. Many studies in Table 1 suffer from long and infrequent follow-up times. Accordingly, some report weak correlations due to high inter-patient variability, and biological expansion of the AE dominating the LET effect [40,42,46,47]. Yang et al. [47] demonstrates this through the rapid progression of mandible osteoradionecrosis in a H&N patient by comparing images taken at 1.5 yrs post-treatment (time of diagnosis) as well as 6 and 11 months thereafter. A correlation between LET and mandible osteoradionecrosis initialisation was observed at the time of diagnosis, whilst AE voxels in later follow-up images were not a result of high LET. Indelicato et al. [51] and Giantsoudi et al. [52] arrived at conflicting conclusions, where a correlation between endothelial cell damage and LET was hypothesised for the former, and no correlation with CNS injury for the latter. Although different AEs were considered between the studies, endothelial cell damage has been associated with radiation-induced CNS injury [57]. The median follow-up time was 24 months [51] for endothelial cell damage, compared to 50.4 months [52] for CNS injury. A possible reason no LET-CNS injury correlation was observed is because any LET dependence was diluted prior to the 50.4 month follow-up. This suggests the need for studies with more frequent follow-up times to reduce the effect of biological progression of the AE and allow at-risk patient groups to be identified for future LET optimisation studies. This is especially important for asymptomatic effects that can go undiagnosed, some of which have already been associated with high LET [38,49].

In scenarios where the treatment plan warrants complicated beam arrangements, the LET distribution becomes highly non-trivial, as will be discussed in the following section. In these situations, the presence of LET hotspots is more likely. Bertolet et al. [40] found that MR image changes in regions of elevated LET were observed in 11 medulloblastoma patients out of a cohort of 26. These same 11 patients had shallower targets with smaller beam angles compared to the rest, resulting in LET hotspots due to use of low energy beams with lower range straggling and a concentration of track-end regions. Similar observations were made in Bolsi et al. [43], where two out of three craniopharyngioma patients who experienced radiation-induced cerebral vasculopathies received an asymmetric beam plan, compared to no patients who received a symmetric plan. This demonstrates the need to consider the nature of the treatment plan, such as beam arrangement, to inform the patient selection protocol for LET optimisation.

**Table 1 cancers-15-04268-t001:** A summary of key findings from clinical studies investigating the correlation between LET and adverse effects. H&N, head and neck; PBS, pencil beam scanning; NR, not reported; LET, linear energy transfer (keV/μm); NTCP, normal tissue complication probability; PS, passively scattered; MR, magnetic resonance; CEBL, contrast-enhancing brain lesions; CTV, clinical target volume; FX, fractions; AE, adverse effects; RICV, radiation-induced cerebral vasculopathies; LGG, low-grade glioma; BD, biological surrogate dose; RIBI, late radiation-induced brain injury; CNS, central nervous system. Dose, LET and BD metrics in this table are defined in Appendix B. The circles represent Red = No; Orange = Maybe; Green = Yes to LET/AE correlation.

Study	Patient Details	Treatment Details	Clinical Endpoint	Follow-Up/Diagnosis (Months)	Key Findings (*p*-Value)
Roberts et al. [58] 	30 Brain (Paediatric)	PBS, ≥50.4 GyRBE	Brain image changes	Median 21 (2–35)	Stronger correlation with BD than with physical prescription dose
Harrabi et al. [38] 	110 LGG (Adult)	NR, Median 54 GyRBE	CEBL lesions	39	Increased incidence of asymptomatic radiation-induced brain injuries with an increased LET${}_{\mathrm{mean}}$.
Bahn et al. [41] 	110 LGG (Adult and Paediatric, AE = 23)	PBS, 45–60 GyRBE	CEBL (total of 67 across all patients)	Median 38 (1–91)	Independent correlation between CEBLs and dose, LET and distance to ventricular system. LET${}_{\mathrm{mean}}$ ranges from 3.56 to 8.18 keV/μm within CEBL regions.
Eulitz et al. [45] 	42 LGG (Adult)	PBS with chemotherapy, 54–60 GyRBE	RIBI (64 in 21 patients)	24	Spatial correlation with RIBI and elevated LET, dose and periventrivular radiosensitivity.
Peeler et al. [39] 	34 Ependymoma(Paediatric)	PS, 54 GyRBE	MR Image Changes	NR	Significant correlation between CEBLs and track-averaged LETmax (>2.5 keV/μm) in CTV (*p* = 0.02). Insignificant change for LETmean (*p* = 0.06), Dmean (*p* = 0.49) and Dmax (*p* = 0.77) in CTV.
Oden et al. [17] 	3 Intracranial	PBS, 50.4–54 GyRBE (28–30 FX), 2 opposed beams	Brain toxicity, Blindness	5–10 (toxicity), 9 (blindness)	Elevated LET2% of 4–6 keV/μm with high dose occurred inside toxicity volumes.
Bertolet et al. [40] 	26 Medulloblastoma (Adult)	PBS, 1.8 GyRBE(30–34 FX)	Brain image changes	17 (2–61)	11 patients showed elevated LET in image change regions of equivalent dose. These patients had shallower targets, used fewer beams and smaller angles.
Bolsi et al. [43] 	16 Craniopharyngioma (Paediatric, AE = 2)	PBS, 54 GyRBE	RICV	14, 24 (for AE patients)	LET${}_{\max}$ and LET${}_{\mathrm{mean}}$ increased significantly (*p* = 0.02) for RICV patients.
Engeseth et al. [44] 	15 Base of Skull (Adult)	NR, 75.6–79.2 GyRBE	Brain image changes	19 (9–33)	LET${}_{\mathrm{mean}}$ and D${}_{\mathrm{mean}}$ increased in image change regions (3.61 keV/μm and 63.5 GyRBE). TD${}_{15}$ ${}^{‡}$ is 63.6 GyRBE and 2 keV/μm or 50.1 GyRBE and 5 keV/μm, respectively.
Yang et al. [53] 	55 Prostate(Paediatric, AE = 9)	PBS, 75.6–79.2 GyRBE(42–44 FX)	Rectal Bleeding	NR	Significant increase in both V (67.8 GyRBE, 2.86 keV/μm) (*p* = 0.007) and V (72.2 GyRBE, 0 keV/μm) (*p* = 0.01) in the rectum for group that experienced rectal bleeding. *
Fossum et al. [59] 	11 H&N	PBS, 60–70 GyRBE	Dermatitis, dry mouth, dysgeusia, fatigue, mucosal infection, oral mucositis, oral and skin pain, pharyngolaryngeal pain, salivary duct inflammation, trismus, weight loss	Immediately post-treatment (3, 6 and 12 mo thereafter)	BD hotspot correlated exactly with AE for 2 patients, correlations strong for AEs in oropharynx and oral cavity, correlations were not as strong in the brain and mandible.
Wang et al. [49] 	203 Breast (Adult, AE = 13)	PBS and PS,50.4 (32–59.4) GyRBE	Low-grade rib fractures	NR	Increase in BD0.5cc in fracture regions by 56.4 Gy.
Giantsoudi et al. [52] 	111 Medulloblastoma (Paediatric)	PS, NR, Distal track-end in brainstem	CNS Injury	50.4	Increase in LETmean for symptomatic AE but no clear correlation.
Indelicato et al. [51] 	73 Ependymoma, 68 Craniopharyngioma, 66 LGG (Paediatric)	NR, 54–59.4 Gy	Brainstem Injury	24	Hypothesised a correlation of high LET with endothelial damage.
Yang et al. [47] 	113 H&N (Adult, AE = 20)	PBS, 1.8–2.1 GyRBE (25–35 FX)	Ulceration, Hemorrhage, Osteoradionecrosis, Mucositis	18, 24, 35	Correlation of Osteoradionecrosis and Mucositis (out-field). No correlation of ulceration and mucositis (in-field).
Sethi et al. [48] 	109 Medulloblastoma (Paediatric)	PS, Median 54 GyRBE	Recurrence	38.8 (1.4–119.2)	No correlation with LET.
Skaarup et al. [56] 	6 Brain (Paediatric)	PS, 36–59.4 GyRBE	Brain image changes	12 post-treatment, 3 thereafter	No strong correlation of image changes with LET as a function of time post-treatment (Due to no significant image changes observed on follow-up scans and small sample size).
Garbacz et al. [42] 	45 Base of Skull (Adult)	NR, 70–74 GyRBE	CEBL	3 post-treatment, 6 thereafter	No clear correlation between CEBL and high LET.
Niemierko et al. [46] 	179 H&N, Brain and Base of Skull (Adult)	PS, ≥59.4 GyRBE	Brain Necrosis	NR	No correlation between LET adjusted for dose and brain necrosis.
Wagenaar et al. [50] 	100 H&N	PBS, 70 GyRBE	Xerostomia, Tube Feeding, Dysphagia	NR	No correlation between LET${}_{\mathrm{mean}}$ nor D${}_{\mathrm{mean}}$ and NTCP for any endpoint due to inter-patient variability.

* V(*x* GyRBE, *y* keV/μm) describes the volume that receives at least *x* GyRBE and *y* keV/µm. ^‡^ Dose/LET resulting in a 15% probability of an adverse effect.

Due to the inverse relation between LET and dose, high LET regions may not always be clinically relevant alone and thus should be considered in conjunction with dose. Bahn et al. [41] studied 110 low-grade glioma patients, and whilst a spatial correlation between contract-enhancing brain lesions and high LET was shown, a correlation between distance from the ventricular system and dose was also observed. Additionally, Yang et al. [53] analysed a cohort of 55 prostate patients to investigate the LET correlation with risk of rectal bleeding. Their analysis quantified LET dependence using the metric $V_{D,L}$ (which represents the cumulative volume receiving a dose of ≥D and LET ≥L). The study found significant increases in V(67.8 GyRBE, 2.86 keV/μm) (*p* = 0.007) and V(72.2 GyRBE, 0 keV/μm) (*p* = 0.01) in the nine patients who experienced rectal bleeding. Similar sentiments are echoed by comparing AE and non-AE CT voxels that receive similar doses [40] or using BD instead of LET [49].

In addition to evidence of clinical benefit, studies such as those in Table 1 may serve to quantify LET-based constraints for future LET optimisation studies or trials, as has become the standard for dose-based optimisations. Hahn et al. [60] attempted to impose constraints derived from clinical data [17,40,41,46,52] by penalising voxels receiving a LET of 2.5 keV/μm with a dose threshold of 40 GyRBE. However, many studies currently choose constraints empirically [17,32,61,62], suggesting that quantifying such constraints is a necessary step for LET optimisation.

In summary, clinical evidence of LET influence on AEs is present. However, limitations such as infrequent follow-up times, inter-patient variability, small sample size and biological expansion of the initial AE tend to mask the extent of its contribution. Although clinical trials on LET optimisation are underway, results are still distant and studies with earlier, more frequent follow-ups are needed to identify LET-caused AEs prior to progression. This would be particularly useful for understanding long-term risks associated with PBT, such as recurrence and secondary cancer. In doing so, LET-based constraints can be better quantified, at-risk patient groups identified, and further biological risk factor evaluations of PBT performed.

### 3.2. LET Optimisation via Delivery Technique

#### 3.2.1. Beam Field Configuration and Spot Size

With novel delivery techniques achieving higher dose conformality (e.g., with PBS), less trivial and higher LET distributions arise. Table 2, Table 3 and Table 4 outline studies that capitalise on beam arrangement to optimise the LET distribution, henceforth referred to as “LET optimisation via delivery technique”. In Figure 2, we show the concentration of high LET in the distal Bragg Peak region when smaller beam angles are used. An increase in AEs has been observed for patients receiving asymmetric PBS PBT plans with smaller beam angles [40,43], suggesting that beam configuration should be considered during patient selection for LET optimisation. The most effective OAR avoidance technique is to carefully select beam angles such that the highest OAR volume possible is spared. This is simple to implement for plans where the dose distribution alone is optimised. However, the situation is complicated when the LET distribution is considered in addition because track-end stopping proximal to critical structures is no longer favourable.

Studies investigating LET changes with beam arrangement have shown that LET${}_{\max}$ (Appendix B) increases significantly as the angle between the two beams approaches 0${}^{\circ }$ due to a build-up of high LET in a small region [10,63,64]. Faught et al. [10] observed an LET${}_{\max}$ increase of 3.3 keV/μm immediately outside the target boundary when a beam angle of 60${}^{\circ }$ is used compared to 180${}^{\circ }$, while inside the target LET${}_{\max}$ increased by 1.2 keV/μm and LET${}_{\mathrm{mean}}$ (Appendix B) remained unchanged. Fjæra et al. [64] made similar observations in that narrowing the angle between two lateral beams increased LET within the brainstem; however, with the addition of a third vertex beam, LET${}_{\mathrm{mean}}$ decreased. It is also found that when the planning target volume (PTV) overlaps with the brainstem, LET${}_{\mathrm{mean}}$ is much lower than if the two structures were juxtaposed. In practice, the achievable LET is also affected by the target size, as demonstrated in Li et al. [65] where LET${}_{\mathrm{mean}}$ decreases with increasing target volume. This highlights the importance of considering both patient anatomy and the details of their treatment plan when deciding whether they are an ideal candidate for LET optimisation.

In addition to patient-specific parameters, PBS PBT is associated with higher LET compared to the PS technique due to lower energy degradation. Typical spot sizes in a PBT clinic range from 2.5 to 12 mm [64,66], where a larger spot size has a smeared, lower LET distribution and less conformal dose. However, it is unclear whether this effect is significant compared to other factors of beam arrangement. Giantsoudi et al. [20] found that otherwise identical plans with a 3 mm spot size resulted in an elevated LET to the target (approximately doubled for some OARs) and lower mean dose to OARs compared to a 12 mm spot size. However, this is contradicted by Fjæra et al. [64], who saw little variance between spot sizes of 3 and 9 mm. Even though this conflict could be affected by the voxel size used, there is scope for further research to quantify the effect of using a small spot size with respect to LET changes.

#### 3.2.2. Beam Multiplicity and Spot-Scanning Proton Arc Therapy

Dose conformality is often improved further by increasing the number of beams. This method is useful for creating conformal dose fields for non-uniform targets at the expense of a larger normal tissue low-dose bath. Additionally, the high modulation of individual beams in normal tissues can lead to more dose hotspots in critical structures. With respect to LET, increasing the beam multiplicity lowers LET${}_{\mathrm{mean}}$ and LET${}_{\max}$ through the voxel averaging effect outside the target boundary but increases each within [17,66]. This infinite beam multiplicity limit is used to motivate spot-scanning proton arc therapy (SPArcT), which assumes that near-perfect dose conformity can be achieved as the beam multiplicity →∞ (i.e., an arc), whilst being more radiobiologically favourable [67]. Its feasibility has been investigated in the literature [61,65,68], but there has been limited clinical use [69]. Table 3 summarises published results of SPArcT with respect to plan evaluation.

Generally, SPArcT or high beam multiplicity plans resulted in a higher volume of tissue receiving low dose and LET [61,65], although dose conformity often improves as more beams are used. This is demonstrated for paediatric brain plans in Toussaint et al. [61]. LET${}_{\mathrm{mean}}$ remained largely unchanged in normal brain tissue, whilst inside the brainstem, the results were inconsistent, increasing in some cases and decreasing in others. However, LET${}_{\max}$ decreased consistently by as much as 3 keV/μm inside the brainstem increasing the number of fields used from 3 to 18. The brain patients analysed in Bertolet et al. [68] and Li et al. [65] saw similar dosimetric changes, but with a decrease in LET${}_{\mathrm{mean}}$ in the brainstem. The prostate patient in Li et al. [65] saw less dosimetric advantage than the brain patients above with an increase in D${}_{\mathrm{mean}}$ (Appendix B) in the rectum and bladder. In comparison, the liver patient examined in the same study saw a decrease in D${}_{\mathrm{mean}}$ by 1.5 GyRBE, despite a larger low-dose bath than a three-beam IMPT plan. These conflicting results suggest that SPArcT may not be favourable for deep-seated tumours such as the prostate, whilst patients with relatively shallow tumours may benefit.

In summary, SPArcT can achieve superior dose and LET target conformity; however, it is at the expense of a higher normal tissue dose, which is important for deeper targets. Further research is needed to understand the impact of low dose normal tissue exposure before the technique can be put into regular clinical use.

#### 3.2.3. Beam Weighting and Shifting

In an attempt to reduce the impact of the distal LET increase in the Bragg Peak, “preferential spot weighting” or “distal-end shifting” outside OARs and inside targets has been investigated [9,17,66,70]. Giantsoudi et al. [70] explored the LET and dose-sparing potential of distal-end shifting beyond the brainstem for posterior fossa tumours. A method to combine the clinical target volume (CTV) and brainstem into a single volume to shift the distal-end beyond the brainstem (LET sparing) was explored, and was compared to the conventional method of aligning the distal edge of the SOBP proximal to the brainstem (dose sparing). Upon comparison of two and three field configurations for each sparing method, the target LET and dose coverage was comparable. However in the brainstem, a significant increase and decrease in D${}_{\mathrm{mean}}$ (*p* < 0.01) and LET${}_{\mathrm{med}}$ (*p* = 0.01), respectively, was observed.

Grassberger et al. [9] considered beam weighting and suggests that after modulation, LET${}_{\mathrm{mean}}$ in the target can range from 1.5 to 4 keV/μm, whereas in critical structures and normal tissue, LET${}_{\mathrm{mean}}$ can exceed 5 keV/μm in areas >70% of the prescription and 8 keV/μm in low dose regions. Guan et al. [8] validated these findings by systematically decreasing the beam weighting of each energy layer toward the distal edge. Both studies suggest that this method can achieve LET sparing in target-adjacent critical structures.

**Table 2 cancers-15-04268-t002:** A summary of the studies that measure changes to the LET distribution via beam arrangement and shifting of the distal track-ends. The metrics in this table are defined in Appendix B. LETOpt, LET-optimised; H&N, head and neck; L/R, left/right; PBS, pencil beam scanning; M/SFO, multi/single-field optimisation; FX, fractions; PTV, planning target volume; RBE, relative biological effectiveness; NR, not reported; PS, passively scattered; CTV, clinical target volume; DE, distal track end.

Study (Intervention)	Patient Details				Treatment Details		Key Findings (*p*-Value)
	Sample Size	Cases	Adjacent or Overlapping OARs		Delivery	Prescription [GyRBE] (Target)	
Henjum et al. [63] ${}^{‡}$ (Beam Configuration + LETOpt)	1	Brain	Brainstem, L optic nerve Rectum, Bladder		PBS, 2 fields (cranial, lateral) ( 34${}^{\circ }$), MFO	54, 30 FX (PTV)	LETOpt resulted in dose sparing of the brainstem and optic nerve, but with little change to LET. ${}^{†}$
	1	Prostate			PBS, 2-bilateral coplanar fields, MFO	67.5, 25 FX (PTV)	LETOpt had little effect on OAR sparing to the rectum and bladder as the fields were directly opposed. ${}^{†}$
	1	H&N	R parotid gland, L pterygoid		3 coplanar fields (50${}^{\circ }$, 30${}^{\circ }$), MFO	50.4, 28 FX (PTV)	LETOpt has little effect on the dose and LET distribution. ${}^{†}$
Fjæra et al. [64] (Beam Configuration)	1	Brain (Paediatric)	Brainstem		PBS, 3 fields (2 lateral, 1 posterior non-coplanar), target shifted to overlap, abut and separate from the brainstem	59.4, 33 FX (PTV)	Addition of a third (vertex) field lowered LET${}_{\mathrm{mean}}$, whilst smaller angles between the 2 lateral fields increased LET${}_{\max}$. No significant LET change when increasing spot size. Shifting the PTV away from the brainstem decreased LET${}_{\mathrm{mean}}$ to the brainstem but increased D${}_{\mathrm{RBE}}$.
Faught et al. [10] (Beam Configuration)	1, 4	Water phantom, Brain (Paediatric)	Brainstem		PBS, two fields at 60, 90, 120 and 180${}^{\circ }$	NR	Inside the target RO margins and use of range shifter had the biggest impact (0.6 keV/μm for both cases). LET${}_{\max}$ increased by 4.3 keV/μm on the outer target boundary as beam angles decreased from 180→60${}^{\circ }$. Lower effect for spot size and SFO vs. MFO.
Giantsoudi et al. [70] (Distal-end Shifting)	6	Ependymoma	Brainstem		PS, 2–3 fields, large/small beam angles, LS= shift DE past brainstem, DS= DE occurs distal to the target and inside brainstem	NR	**Brainstem:** From DS to LS for both 2 and 3 fields, D${}_{\mathrm{mean}}$ increased (*p* < 0.01, <0.01), D${}_{\max}$ decreased (*p* = 0.04, 0.02) (<1 GyRBE) and LET${}_{\mathrm{med}}$ decreased (*p* < 0.01, =0.01). **Brainstem-CTV overlap:** No statistically significant change (*p* > 0.05) in D${}_{\max}$ for LS from 3 to 2 fields. Significant decrease (*p* = 0.04) in D${}_{\mathrm{mean}}$ and LET${}_{\mathrm{med}}$ (of only 0.1 keV/μm).

^†^ Information extracted from figures as not reported in the text. ^‡^ This study analyses the beam arrangement and its impact on the inverse planning LET optimisation approach discussed in Section 3.3.

**Table 3 cancers-15-04268-t003:** A summary of the studies that measure changes to the LET distribution by increasing the number of beams (spot-scanning proton arc therapy). The metrics in this table are defined in Appendix B. SPArcT, spot-scanning proton arc therapy; IMPT, intensity-modulated proton therapy; N/A, not applicable.

Study (Intervention)	Patient Details				Treatment Details		Key Findings (*p*-Value)
	Sample Size	Cases	Adjacent or Overlapping OARs		Delivery	Prescription [GyRBE] (Target)	
Li et al. [65] (SPArcT + 2-stepLETOpt)	1	Liver	Normal liver tissue		IMPT 2 fields vs. SPArcT	75, 25 FX (CTV)	**CTV:** From 2 field IMPT→SPArcT both with LETOpt, dose conformality similar and LET${}_{\mathrm{mean}}$ increased from 2.4 to 4.9 keV/μm. **Normal liver tissue:** From 2 field IMPT→SPArcT with LETOpt, the low dose bath is larger but D${}_{\mathrm{mean}}$ decreased by 1.5 GyRBE.
	1	Prostate	Rectum		IMPT 2–8 fields vs. SPArcT	78, 39 FX (CTV)	**CTV:** From 2 field IMPT→SPArcT, similar dose coverage, LET${}_{\mathrm{mean}}$ increased from 4.38 to 5.06 keV/μm. D${}_{\mathrm{mean}}$ to the **bladder and rectum** increased.${}^{†}$
	1	Brain	Brainstem, Chiasm, L and R optic nerve		IMPT 3 fields vs. SPArcT	54, 30 FX (CTV)	**CTV:** 3 field IMPT→SPArcT both with LETOpt saw similar dose coverage and LET${}_{\mathrm{mean}}$ increased from 3.13 to 4.03 keV/μm. **OAR:** Smaller D${}_{\mathrm{mean}}$ for brainstem and chiasm but higher for optic nerve${}^{†}$. LET${}_{\mathrm{mean}}$ decreased from 2.74 to 2.14 keV/μm, 3.45–2.43 keV/μm, 4.09–2.96 keV/μm and 3.22–2.66 keV/μm for the brainstem, chiasm, L and R optic nerve, respectively.
Bertolet et al. [68] (SPArcT)	1	Water Phantom (Cylinder)	N/A		Compared SPArcT to 2–3 field IMPT (coplanar or noncoplanar)	2, 1 FX (N/A)	From 3 field IMPT→SPArcT, LET${}_{\mathrm{mean}}$ and LET${}_{\max}$ increased in the target. In the brainstem LET${}_{\max}$ decreases by ≈50% for SPArcT vs. 2 beam IMPT but D${}_{\mathrm{mean}}$ increases.
	3	Brain	Brainstem (All), R optic nerve (1), L hippocampus (1)			26, 30 FX (PTV)	**CTV:** In all cases D${}_{2\%}$ and D${}_{\mathrm{mean}}$ did not change but LET${}_{98\%}$, LET${}_{\mathrm{mean}}$ and LET${}_{2\%}$ nearly doubled. **Brainstem:** D${}_{2\%}$ halved for patient 1 but did not change for the others. D${}_{\mathrm{mean}}$ increased slightly for patients 2 and 3 and did not change for 1. D${}_{98\%}$ increased significantly for all patients.
Toussaint et al. [61] (SPArcT)	4	Craniopharyngioma (Paediatric)	Brainstem, Temporal Lobes		PBS, 3–36 fields (coplanar, sagittal)	54, 30 FX (NR)	Dose and LET conformality improved as beam multiplicity increased. Volume that received >3.5 keV/μm decreased but at expense of increased volume receiving <3.5 keV/μm. Low isodose volume increased with beam multiplicity.

^†^ No quantitative data given, read from figures.

**Table 4 cancers-15-04268-t004:** A summary of the studies that measure changes to the LET distribution using the LET-painting technique. The metrics in this table are defined in Appendix B. STP, split-target plan; FTP, full-target plan; #-S/FTP, [Number of beams]-S/FTP; LP, LET painting; DP, Dose painting; SOBP, spread-out Bragg Peak.

Study (Intervention)	Patient Details				Treatment Details		Key Findings (*p*-Value)
	Sample Size	Cases	Adjacent or Overlapping OARs		Delivery	Prescription [GyRBE] (Target)	
Fager et al. [71] (LET painting)	8	Prostate	Rectum, Bladder		PBS, 2–7 beams, target was split so all beams stopped at its centre (STP) vs. full target plan (FTP)	79.2, 44 FX	**CTV:** LET${}_{\mathrm{mean}}$ 2.5 keV/μm (FTP), increased by 1.5 (*p* = 0.008), 1.8 (*p* = 0.016) and 2.1 keV/μm (*p* = 0.031) for 2STP, 4STP and 7STP, respectively. **Non-OAR normal tissue:** LET${}_{\mathrm{mean}}$ 2.8 keV/μm (FTP), decreased by 0.1 (*p* = 0.125), 0.5 (*p* = 0.016) and 0.5 keV/μm (*p* = 0.031) for 2STP, 4STP and 7STP, respectively. **Rectum:** LET${}_{\mathrm{mean}}$ 2.9 keV/μm for FTP, increased by 1.5 keV/μm (*p* = 0.008) for 2STP and decreased by 0.1 keV/μm for 4STP and 7STP (*p* = 0.81, 0.84), respectively. **Bladder:** LET${}_{\mathrm{mean}}$ 3.2 keV/μm for FTP, increased by 0.5, 0.0 and 0.2 keV/μm for 2STP, 4STP and 7STP, respectively. D${}_{90\%}$ decreased by 5.3, 4.4 and 4.4 GyRBE for 2STP, 4STP and 7STP, respectively.
Guan et al. [8] (LET-Painting)	1	Water Phantom	N/A		PBS, 2 laterally opposed fields, with flat SOBPs (Case A) and decreased spot weighting in SOBPs toward the distal-edge (Case B)	NR	**Case A:** LET inside the target 5.6–5.7 keV/μm on the target edge and 2.6 keV/μm in the centre. LET outside the target <1 keV/μm.${}^{†}$ **Case B:** LET inside the target is 4.3–4.4 keV/μm on the distal edges and 3.8 keV/μm in the centre. Outside the target, LET is <1 keV/μm.${}^{†}$
Malinen et al. [72] (LET-Painting)	1	Tumour Model with hypoxic regions	N/A		NR, no field information, compared dose painting (DP), LET painting (LP) and DP+LP to reference plans (no DP or LP)	NR, up to 16 FX	LET range 1–11 keV/μm. Therapeutic ratio for DP = 1.43, LP = 1.09 and DP + LP = 1.45

^†^ No quantitative data given, read from figures.

While beam weighting considering both conventional dose and LET optimisation has been thoroughly explored, the process is dependent on the shape and location of the target [64,65]. By implementing LET-based objectives and constraints into the objective function of the treatment plan, the process can be more “automated” compared to the methods discussed thus far. This “LET optimisation via the objective function” will be discussed in the following section.

#### 3.2.4. LET Painting

The studies discussed in Section 3.2 thus far have aimed to quantify the effect of LET when beam angles, multiplicities and weightings are varied systematically. *LET painting* is a method whereby all these properties are used to manipulate the LET distribution within the target, or alternatively to “paint” the target with LET. The method is applied to either the whole target volume [8,71,72] or to smaller targets within [71]. The PBT results from these studies are summarised in Table 4.

Some studies [64,65] have shown that the achievable LET inside the target is dependent on its size and shape. Smaller volumes are able to better concentrate high LET than larger ones [65] and non-uniform targets require more modulated beams resulting in LET hotspots being diluted by low LET regions of opposing beams [63]. Malinen et al. [72] investigated LET painting in hypoxic regions within the tumour and achieved a LET range of 1−11 keV/μm by concentrating LET only within voxels corresponding to hypoxic regions within the target. Alternatively, one can consider the full volume by splitting the target into segments such that the beams stop at its centre [71,73,74]. For Fager et al. [71], this boosted LET${}_{\max}$ from 3 to 5 keV/μm, increasing further for four or seven beam configurations. The method was applied to prostate plans, therefore increasing the number of beams also resulted in a significant dose increase to the bladder and rectum.

As discussed in the previous section, a method proposed by Guan et al. [8] was able to apply preferential spot weighting to avoid high LET regions at the penumbra of the beam profile without the need for additional beams. This was achieved by systematically decreasing the weight of each energy layer to effectively zero in the distal edge of each SOBP. They found that two flat (i.e., equally weighted) opposed beams achieved an LET${}_{\max}$ of 5.6 keV/μm inside the target boundary, whereas the same configuration with systematically decreasing beam weighting for high energy layers yielded an LET${}_{\max}$ of 4.4 keV/μm. Whilst this decreases LET inside the target, the high LET region is located more centrally in the target, thus improving plan robustness compared to when high LET regions are close to the boundary.

LET painting is a computationally cheap optimisation alternative. Its region-selective nature is suitable in cases where hypoxia is common (e.g., pancreatic cancer [75]) and high LET radiation is ideal. Instances where high LET to the whole volume is beneficial can use the split target approach to shift LET hotspots well within the target boundaries. This approach improves plan robustness compared to those that shift LET just within the boundary. One drawback of the LET painting method is that its main objective is LET escalation within the target; therefore, one should exercise caution in situations where LET sparing in a critical structure is a priority.

### 3.3. Objective Function-Based LET Optimisation

#### 3.3.1. Selection of Constraints, Objectives and Algorithms

The LET optimisation methods discussed thus far use a “LET optimisation via delivery technique” approach, whereby beam properties are directly exploited to spare critical structures and boost LET within the target. Whilst these methods are computationally cheap, the high degeneracy of LET with dose makes the finding most optimal plan difficult, and labour-intensive [9,74]. To ease the workload associated with LET optimisation via delivery technique, inverse planning approaches have been proposed where LET objectives and constraints are integrated with the dose objective function to ensure the most mathematically optimal plan is delivered.

Most dose-based objective functions consist of terms that ensure uniform dose with high conformity at the target prescription dose, minimise dose to OARs or below a constraint and sometimes minimise dose to non-OAR normal tissues [76]. These objectives and constraints can either be hard or soft (i.e., allow some discrepancy) and are based on dose to all, or part of, the volume (e.g., D${}_{\mathrm{mean}}$ of *X* to *Y*% of the volume). When implementing LET optimisation, the optimal beam configuration *x* is chosen by minimising the objective function:(2)$$F\left(x\right)=\min\left(\overset{1-\mathrm{step}}{\overset{︷}{\underset{2-\mathrm{step}\;\left(1\right)}{\underset{︸}{F_{D}\left(x\right)}}+\underset{2-\mathrm{step}\;\left(2\right)}{\underset{︸}{F_{L}\left(x\right)}}}}\right)\;.$$
The computation can be performed by minimising $F_{D}\left(x\right)$ (dose objective function) and $F_{L}\left(x\right)$ (LET objective function) simultaneously (1-step) [20,31,32,34,77,78,79] or minimising and fixing $F_{D}\left(x\right)$ to use as a warm start for $F_{L}\left(x\right)$ (2-step) [30,60,65].

Due to differences between studies in objectives, patients and reported metrics, a fully quantitative meta-analysis could not be performed. Instead, Table 5 summarises the studies, with respect to how often the objectives are met, eluding to any therapeutically favourable methods. Figure 3 outlines the differences between the dose and LET objectives included in Equation (Equation 2) for each study in Table 5. In every study, LET inside the OARs is included in Equation (Equation 2) as a quantitative constraint or general objective. All but four studies include a LET-based objective inside the target. The result is that any LET hotspots close to the target are likely to be pushed within the target boundary. However, if the OAR is not close to the target, the LET hotspot is likely pushed into normal tissue. Thus far, no studies have included an objective to keep LET out of non-OAR normal tissue, which may be beneficial in preventing AEs, such as brain image changes; see Section 3.1.

Currently, most studies use quadratic dose and LET, BD or LET-adjacent objectives [30,31,34,60,65,79], while others use linear [78,79] and nonlinear objectives [32,77]. Cao et al. [78] and Chen et al. [79] applied the same 1-step linear objective function to five brain (Glioblastoma, Anaplastic Astrocytoma and three Ependymoma) and 10 prostate cases, respectively. Their objectives are to maintain identical or improved target dose coverage, whilst maximising and minimising LET to the target and OARs, respectively. Unlike Cao et al. [78], the number of fields was increased from two to eight in Chen et al. [79]. The plans remained dosimetrically identical, regardless of the inclusion of LET constraints. However, increases of 0.4–0.5 keV/μm in the GTV LET${}_{99\%}$ (Appendix B) for two of three ependymoma patients were observed in Cao et al. [78]. Similarly, Chen et al. [79] saw a 1 keV/μm decrease of LET${}_{98\%}$ in the CTV across all prostate patients (*p* < 0.05), for two to eight field configurations, suggesting the target LET objective was met in most cases. In Cao et al. [78], the brainstem and chiasm abutted or overlapped with the CTV, for which LET${}_{\mathrm{mean}}$ remained mostly unchanged aside from one patient. LET${}_{\max}$ decreased in the brainstem for all cases between 0.2 and 5.8 keV/μm. The large range is likely due to inter-patient differences in target location. From the information that was provided in each study, and the linear objective function achieved its dose and LET objectives; however, the decrease in LET is extremely patient-dependent. Additionally, for both studies, the LET alone was assessed for changes. As dose remained largely unchanged between the dose and LET-optimised plans, it is possible that the high LET regions were associated with clinically negligible doses, suggesting that linear constraints do not accurately reflect the dose and LET relationship.

A majority of studies in Table 5 use quadratic constraints. Bai et al. [31] analysed a similar cohort of patients to Cao et al. [78], with similar objectives as quadratic constraints instead of linear. Although D${}_{\mathrm{mean}}$ in the CTV did not vary from the reference plan, D${}_{2\%}$ (Appendix B) increased slightly (up to 1 GyRBE), and LET${}_{2\%}$ increased (up to 1.35 keV/μm) for three of five patients. In the brainstem, D${}_{\mathrm{mean}}$ increased from the reference plans for all patients, whilst D${}_{2\%}$ remained unchanged. However, LET${}_{2\%}$ and LET${}_{\mathrm{mean}}$ decreased significantly from the reference plans. Similarly, Bai et al. [31] (Chapter 3) also loosely achieved their objectives, but the overall LET distribution was lower in the brainstem for all patients. *Therefore, based on currently published studies, quadratic constraints are generally more representative of LET and dose than linear constraints*.

Most studies perform the optimisation using Quasi–Newton (QN) or interior-point methods [81], where the derivative of Equation (Equation 2) is solved in full for each iteration and, as such, the nonlinearity of the LET term $F_{L}\left(x\right)$ in Equation (Equation 2) is not accounted for. The recent method proposed by Li et al. [77] uses an iterative convex relaxation (ICR) method [81] to iteratively solve the derivative of Equation (Equation 2), whilst assuming $F_{L}\left(x\right)$ to be nonlinear. The study compared the QN and ICR methods, with and without the $F_{L}\left(x\right)$ term (henceforth QN, ICR, QN-LET and ICR-LET, respectively), for one lung, abdominal, H&N and brain cases. The results showed a marginal improvement of LET${}_{\mathrm{mean}}$ in the OARs (heart, left parotid, brainstem and bowel for the lung, H&N, brain and abdominal case, respectively) between QN and QN-LET, but was reduced further with ICR-LET for all but the H&N case. In the 1 cm boundary around the CTV, LET${}_{\mathrm{mean}}$ remained unchanged for QN, QN-LET and ICR but decreased with ICR-LET for all cases. Compared to the other methods, ICR-LET also achieved a substantial decrease and increase in D${}_{\mathrm{mean}}$ in the OARs and target, respectively. Whilst an extensive plan evaluation was not performed in this study, it suggests that nonlinear constraints and the ICR method are more effective at optimising clinically relevant LET, as well as accounting for the inverse relationship between LET and dose.

Based on current evidence, nonlinear objectives perform best at managing the LET distribution in addition to producing clinically acceptable plans for a wide range of cases. A systematic analysis of linear, quadratic and nonlinear constraints on the same patient cohort would allow a more certain conclusion to be drawn as to which is the most favourable.

#### 3.3.2. Managing the Dose–LET Trade-Off

A simple way to account for the inverse dose–LET relationship is to optimise over BD, Equation (Equation 1), in place of LET-only objectives [30,31,32,33,34,35]. Although the method requires assumption of a scaling parameter *c* (μm/keV), it has been shown to reduce variability in treatment planning [18]. Unkelbach et al. [30] retrospectively planned five brain and base of skull cases with a 2-step method for comparison with purely dose-optimised reference plans. In this optimisation, BD was only minimised in the OARs, while purely dose-based objectives were used in the target. The BD-optimised plans were dosimetrically comparable to the reference plans in the PTV, which is expected as the dose-based objectives did not change. However, BD was reduced for all cases by up to 52% in the abutting brainstem. Bai et al. [34] used a similar approach, but included an objective to maximise BD in the target for a cohort of four brain and H&N patients. The plans were also dosimetrically comparable; however, BD${}_{98\%}$ (Appendix B) increased by 17% in the CTV and BD${}_{\mathrm{mean}}$ (Appendix B) decreased by 23% in the brainstem, thereby achieving its objectives.

When LET objectives are used, dose threshold limits can be imposed to ensure only clinically relevant LET voxels are optimised. By considering all LET voxels with no dose threshold, we steer the overall LET distribution toward higher values [82]. Hahn et al. [60] proposed an alternative 2-step method wherein only voxels with a dose exceeding the threshold are included in the LET optimisation. Ten cranial patient plans, where the CTV overlaps with the brainstem and optical nerves, were assigned a voxel threshold of LET > 2.5 keV/μm where *D* > 40 GyRBE, the values for which are derived from clinical data [17,40,41,46,52]. BD objectives were not defined for the CTV and thus LET optimisation had no dosimetric effect on the target. However, LET${}_{\max}$ was reduced by 19% in the OARs and normal brain tissue D${}_{\mathrm{mean}}$ increased by <3%, indicating that the track-ends were shifted out of the brainstem and into non-OAR normal tissue.

To ensure results such as those in Table 5 are not susceptible to bias in favour of LET optimisation, only clinically relevant LET should be considered by optimising over BD or imposing dose thresholds on LET voxels. This would ensure that computing resources are spent on optimising voxels that will enhance the therapeutic effect within a reasonable computation time, hence becoming more clinically tractable than LET-only approaches.

#### 3.3.3. 1-Step versus 2-Step Optimisations

Thus far, a number of 1- and 2-step LET optimisation methods have been discussed. The results in Table 5 shows that each study generally achieves their objectives in Figure 3 to varying degrees. As discussed above, the success of each method is extremely dependent on patient anatomy and clinical target. In a retrospective study, Gu et al. [32] performed a comparison between 1- and 2-step methods on the same cohort of H&N and base of skull patients. The optimisation was performed over both beam weights and angles (the studies discussed until now have optimised over beam weights only), where high BD was encouraged in the target and minimised in the OARs. Both methods maintained equivalent PTV dose coverage with no stand-out method in regard to BD${}_{\max}$ (Appendix B) and BD${}_{\mathrm{mean}}$. However, BD${}_{\min}$ (Appendix B) in the PTV increased for all LET-optimised methods. In the PTV-adjacent OARs, both methods were able to reduce D${}_{\mathrm{mean}}$ and D${}_{\max}$ (Appendix B). Generally, the 1-step method reduced BD${}_{\max}$ and BD${}_{\mathrm{mean}}$ more than the 2-step method.

Based on current evidence, it is unclear which method is the stand-out, since both generally achieved their objectives. With regard to clinical implementation, the 2-step method would be most compatible. Conceptually, the high degeneracy of LET and dose may suggest that optimising the two distributions simultaneously (1-step) will lead to a more mathematically optimal plan; however, this approach can take longer to compute due to the larger parameter space. The 2-step method allows the planner to first assess the dose distribution, compute the LET distribution and use this to decide whether the patient would be a good candidate for LET optimisation. This would reduce the workload required to implement LET optimisation clinically.

#### 3.3.4. Optimising over Beam Weights and Angles

Of the studies summarised in Table 5 and Figure 3, most optimise over the beam weights only, whereas two also include beam angles [31,32]. This is despite the strong influence of beam angle and multiplicity on the LET distribution. Often, beam angles are pre-selected and in early PBT treatments, beam angles stopping proximal to critical structures were seen as dosimetrically favourable, with little consideration given to LET. However, recently, the unaccounted for LET effect is minorly considered, particularly for serial organs such as the brainstem and spinal cord. Beams stopping proximal to serial organs is a primary cause of increased biological effect inside critical structures.

Gu et al. [32] included beam angle as an optimisable parameter to compare with pre-selected beams. It was found that when the optimiser is allowed to choose the beam angles, the gantry and couch angles changed either as little as 2${}^{\circ }$ to over 100${}^{\circ }$ from the pre-selected beam angles across both skull base and H&N cohorts. Bai et al. [31] made similar observations for a water phantom and two brain patients. Here, only the gantry angle was optimised, resulting in shifts between dose-optimised, beam angle dose-optimised, LET-optimised and beam angle LET-optimised plans as much as 10${}^{\circ }$ to 40${}^{\circ }$, respectively.

Gu et al. [32] observed larger BD reductions in PTV-adjacent OARs compared to outer-field OARs, whereas the outer-field OARs dosimetrically benefited more often. This change is likely a direct result of the shifted beam angles. Similarly, Bai et al. [31], who only analysed CTV-adjacent OARs, saw little dosimetric benefit, whilst BD${}_{98\%}$ and BD${}_{2\%}$ increased and decreased in the CTV and OAR, respectively. Therefore, incorporating beam angle selection will see more dosimetric benefit in out-of-field OARs and biological effect benefit for target-abutting OARs. One drawback is that optimising over both beam weights and angles will likely increase the computation time significantly. Although this is addressed in current studies by limiting the number of beam candidates available for selection by the optimiser [31,32], the required computation time may not be justifiable for current clinical implementation. However, as Section 3.1 has shown, optimising beam angles with LET in mind may reduce adverse effects in patients with highly asymmetric treatment plans, such as brain cases.

#### 3.3.5. Track-End Objectives as a LET Surrogate

A major drawback of LET optimisation is the long computation time required to calculate and iteratively vary the LET distribution during optimisation. Times ranging from 20 min to 2 h, depending on the complexity of the plan and optimisation method [32,34,77], are among the biggest concerns in making the method clinically tractable. Using the direct correlation between LET and proton track-ends, methods that optimise over the number of track-ends in place of LET have been proposed [17,34,60,62,70]. Track-end objectives can be implemented into the objective function the same way as LET or BD objectives, where the number of tracks stopping inside a volume are optimised.

One of the first attempts at this method by Traneus et al. [62] retrospectively re-planned six intracranial and H&N patients, with OARs overlapping the PTV. Track-end constraints for OARs were chosen empirically, whilst the target volume was only dose-optimised. For all intracranial patients, LET${}_{\mathrm{mean}}$ and LET${}_{\max}$ (computed only for voxels with dose >5% of the prescription) decreased in the brainstem, chiasm, optical nerves, pituitary gland and normal brain tissue. In the CTV, both LET${}_{\mathrm{mean}}$ and LET${}_{\max}$ increased despite the exclusion of a target track-end objective. The same behaviour was observed for the H&N patients, with the exception of the target boundary, for which LET${}_{\mathrm{mean}}$ and LET${}_{\max}$ did not change. However, there were only small dosimetric changes, no higher than 0.5 GyRBE, across all plans. There are, however, exceptions for some OARs, likely due to inter-patient variability.

Oden et al. [17] applied an identical objective function to three intracranial patients with PTV-overlapping OARs. The LET-optimised plans used a multi-field optimisation (MFO, with both two and three beams), whereas the reference plan used single-field optimisation (SFO). All plans were dosimetrically equivalent in the CTV, with only LET${}_{2\%}$ increasing by <1 keV/μm. LET${}_{\mathrm{mean}}$ and LET${}_{2\%}$ decreased significantly for all overlapping OARs between dose- and dose-track-end-optimised plans with the same number of beams. The addition of a third beam was explored to understand its effect on track-end optimisation. It was found that the OAR LET decrease was consistently larger for three-field plans compared to two due to the increased degrees of freedom available to the optimiser and overall lower LET distribution. Similarly, Hahn et al. [60] and Bai et al. [34] achieved all their objectives using track-ends as effectively as the compared LET-optimised plans.

Overall, track-end optimisation is an attractive alternative to LET optimisation. It is able to produce equivalent plans to LET optimisation in less time, as it does not need to compute the LET distribution, where most of the additional computation time is spent [32]. Instead, it is simply the number of track-ends per volume required for each iteration. For example, Bai et al. [34] reported a computation speed-up of approximately 30% between track-end and LET-optimised plans, which is a significant improvement, whilst also producing clinically acceptable plans.

**Table 5 cancers-15-04268-t005:** A summary of published LET optimisation methods via an objective function. Unless stated otherwise, the reported data are a comparison between plans that are optimised with dose-only and with LET. The corresponding objective function details are presented in Figure 3. The metrics in this table are defined in Appendix B. RO, Robust Optimisation; LET-based RO, LETRO; OAR, organs at risk; CTV, clinical target volume; PBS, pencil beam scanning; FX, fractions; DoseOpt, dose-based optimisation; LETOpt, LET-based optimisation; NR, not reported; H&N, head and neck; BD, biological surrogate dose; GTV, gross target volume; QN, quasi-newton; ICR, iterative convex relaxation; PTV, planning target volume; OAR, organ at risk; TEOpt, track-end optimisation; DDOpt, dirty dose optimisation.

Study (Intervention)	Patient Details				Treatment Details		Key Findings (*p*-Value)
	Sample Size	Cases	Adjacent or Overlapping OARs		Delivery	Prescription [GyRBE] (Target)	
Bai et al. [31] (1-step) (Ch. 3)	5	Glioblastoma, Anaplastic Astrocytoma, Ependymoma	Brainstem, Optical Chiasm (CTV)		PBS, 3–4 coplanar and non-coplanar fields	54–59.2, 30–32 FX (CTV)	**CTV:** D${}_{98\%}$ remained unchanged from DoseOpt to LETOpt. D${}_{2\%}$ increased for all cases by <1 GyRBE. LET${}_{98\%}$ increased by up to 0.47 keV/μm for all but 1 case. **Brainstem: ** D${}_{\mathrm{mean}}$ increased in all cases up to 2.89 GyRBE from DoseOpt→LETOpt. D${}_{2\%}$ either increased or remained the same. LET${}_{\mathrm{mean}}$ and LET${}_{2\%}$ reduced significantly for all cases.
Li et al. [77] (1-step)	1	Lung	Heart		PBS, 3 coplanar fields ${}^{†}$	54, 30 FX (CTV)	From QN→ICR methods ± LETOpt: **1 cm CTV Boundary:** D${}_{\mathrm{mean}}$ decreased by <1 GyRBE, LET${}_{\mathrm{mean}}$ decreased by 0.5 keV/μm (for ICR more than QN) and BD${}_{\mathrm{mean}}$ decreased by <1 Gy. **Heart:** Insignificant decrease in D${}_{\mathrm{mean}}$. Decreases of <0.5 keV/μm and <1 Gy in LET${}_{\mathrm{mean}}$ and BD${}_{\mathrm{mean}}$, respectively. **Computation time:** 3-fold decrease from QN→ICR ± LETOpt.
	1	H&N	L parotid gland		PBS, 2 coplanar fields ${}^{†}$	63.6, 33 FX (CTV)	**1 cm CTV Boundary:** Same as lung case except LET${}_{\mathrm{mean}}$ decreased by 0.6 and 0.5 keV/μm for ICR and QN, respectively. **L parotid gland: ** BD${}_{\mathrm{mean}}$ decreased by ≈2 Gy for ICR vs. <1 Gy for QN. **Computation time:** 9 fold decrease from QN→ICR ± LETOpt.
	1	Abdominal	Bowel		PBS, 2 coplanar fields ${}^{†}$	50, 25 FX (CTV)	**1 cm CTV Boundary: ** Similar trend to lung case. **Bowel:** Same trend as lung case.
	1	Brain	Brainstem		PBS, 2 coplanar fields ${}^{†}$	54, 30 FX (CTV)	**1 cm CTV Boundary:** Same as lung case except LET${}_{\mathrm{mean}}$ decreased by 0.57 and 0.59 keV/μm for QN and ICR, respectively. **Brainstem: ** D${}_{\mathrm{mean}}$ increased by 1.5 and 0.5 GyRBE for ICR and QN, respectively. LET${}_{\mathrm{mean}}$ decreased by <0.5 keV/μm for both cases. BD${}_{\mathrm{mean}}$ increased by 1.5 Gy and decreased by 1.2 Gy for QN and ICR, respectively. **Computation time:** 6 and 2-fold decrease from QN→ICR ± LETOpt, respectively.
Chen et al. [79] (1-step)	10	Prostate	Rectum ${}^{†}$		PBS, 2, 4, 6 and 8 coplanar fields	78, 39 FX (NR)	**CTV:** LET${}_{\mathrm{mean}}$ and LET${}_{98\%}$ increased significantly by 53–63% (*p* < 0.05) and 63–70% (*p* < 0.05), respectively.
Giantsoudi et al. [20] (1-step)	2	Pancreas	NR		PBS, NR	25, 5 FX	LET${}_{\mathrm{mean}}$ variation over base plans in Pareto-space for small vs. large spot size for patients 1 and 2. **PTV:** Smaller variation (7.9, 14.3% vs. 1.2, 5.1%) **Liver:** Same trend as PTV with 61.3, 38.1% vs. 11.7, 13.9%.
	5	H&N	Brainstem (PTV) ${}^{†}$		PBS, NR	50.4-59.4, 28–33 FX	Similar trend in PTV to pancreatic patients except LET${}_{\mathrm{mean}}$ variation in brainstem up to 222% for small spot size.
Cao et al. [78] (1-step)	5	Brain (Glioblastoma, Anaplastic Astrocytoma, Ependymoma)	Brainstem, Chiasm (CTV)		PBS, 2–3 noncoplanar beams	48–54, 28–30 FX (GTV)	**GTV:** Negligible changes in D${}_{1\%}$ and D${}_{99\%}$, increases in LET${}_{1\%}$ and LET${}_{99\%}$ by ≈1 keV/μm for 3/5 cases and increases of 2–3 keV/μm for the rest. **Brainstem:** No change to D${}_{\max}$ and D${}_{0.1cc}$, LET${}_{\max}$ decreased by up to 2.5 keV/μm in 3 cases with no change to the rest, LET${}_{0.1cc}$ decreased by up to 1.5 keV/μm in 2/5 cases. **Chiasm:** No change in D${}_{\max}$ and D${}_{0.1cc}$, large decrease in LET${}_{\max}$ and LET${}_{0.1cc}$ for 1/5 cases.
Gu et al. [32] (1- and 2-step)	3	Base of Skull	Chiasm, Optic Nerve (PTV) ${}^{†}$		PBS, 2–4 non-coplanar beams (selected from 600 candidate beams)	56–74, NR (CTV)	**PTV:** D${}_{\max}$, D${}_{95\%}$, BD${}_{\mathrm{mean}}$, BD${}_{\max}$ unchanged with LETOpt±angle selection. BD${}_{\min}$ increased for all LETOpt plans. ${}^{†}$ **In-field OARs:** BD${}_{\mathrm{mean}}$, BD${}_{\max}$ decreased the most for LETOpt with angle selection. D${}_{\max}$ and D${}_{\mathrm{mean}}$ also decreased by up to 3 GyRBE with LETOpt+angle selection. ${}^{†}$ **Out-field OARs:** Smaller decreases in BD${}_{\mathrm{mean}}$, BD${}_{\max}$ were observed compared to in-field OARs. Larger decreases in D${}_{\mathrm{mean}}$, D${}_{\max}$ compared to in-field OARs. ${}^{†}$ **1 vs. 2-step:** Dosimetrically equivalent, BD${}_{\mathrm{mean}}$ and BD${}_{\max}$ lowered more for 1-step than 2-step.${}^{†}$
	3	H&N	NR			54–63, NR (CTV)	**PTV:** Same as Base of Skull **OARs:** Same as Base of Skull except D${}_{\mathrm{mean}}$ increased by 7–8 GyRBE for 1 patient. ${}^{†}$
Bai et al. [34] &Bai et al. [31] (Ch. 6) (1-step)	2	Brain	Brainstem		PBS, 3–4 coplanar or noncoplanar (out of 36 candidates)	54, 30 FX (CTV)	**CTV:** From DoseOpt→LETOpt ± angle selection, D${}_{98\%}$ did not change and D${}_{2\%}$ increased by ≤0.4 GyRBE. BD${}_{98\%}$ and BD${}_{2\%}$ increased by 2 and 3 Gy, respectively. **Brainstem:** D${}_{2\%}$ and D${}_{\mathrm{mean}}$ increased by <0.5 GyRBE. BD${}_{2\%}$ mostly unchanged for 1 case and decreased by 3 Gy for the other. BD${}_{\mathrm{mean}}$ decreased for both cases. **Beam Angle Selection:** 2 of the 3 fields differed by 30${}^{\circ }$ with LETOpt.
	2	H&N	NR		PBS, 3 coplanar or noncoplanar (out of 36 candidates)	60 GyRBE, 30 FX (NR)	**CTV:** From DoseOpt→LETOpt ± angle selection, with no change in D${}_{98\%}$, D${}_{2\%}$, BD${}_{98\%}$ and BD${}_{2\%}$. **OARs:** Small decreases of D${}_{2\%}$, D${}_{98\%}$, BD${}_{2\%}$ and BD${}_{98\%}$ within 1 GyRBE and 1 Gy. Changes were not as significant as brain patients.
Traneus et al. [62] (1-step)	3	H&N	Pituitary Gland ${}^{†}$ (PTV)		PBS, 3 coplanar	56–70, 35 FX (PTV)	**CTV:** D${}_{\mathrm{mean}}$, D${}_{2\%}$ unchanged, decreases of ≤0.7 keV/μm and ≤0.5 keV/μm in LET${}_{2\%}$ and LET${}_{\mathrm{mean}}$, respectively. **Pituitary Gland:** D${}_{\mathrm{mean}}$ unchanged, D${}_{2\%}$ decrease <0.5 GyRBE. Decreases of ≤0.5 and 1–3 keV/μm in LET${}_{2\%}$ and LET${}_{\mathrm{mean}}$.
	3	Intracranial	Brainstem, Chiasm ${}^{†}$ (PTV)		PBS, 2 coplanar or noncoplanar	54, 30 FX (PTV)	**CTV:** D${}_{\mathrm{mean}}$ and D${}_{2\%}$ unchanged. Increase of 1–2 and 0.5 keV/μm for LET${}_{2\%}$ and LET${}_{\mathrm{mean}}$, respectively. **Brain-CTV:** D${}_{\mathrm{mean}}$, D${}_{2\%}$ unchanged. Change of <0.2 and <0.6 keV/μm in LET${}_{\mathrm{mean}}$ and LET${}_{2\%}$. **OARs:** D${}_{\mathrm{mean}}$, D${}_{2\%}$ increase and decrease of <0.8 GyRBE in chiasm and brainstem, respectively. LET${}_{\mathrm{mean}}$ decreased by 1–3.4 keV/μm and LET${}_{2\%}$ decreased by 1–3 keV/μm in both the brainstem and chiasm.
Oden et al. [17] (1-step)	3	Intracranial	Brainstem, Chiasm, Optic Nerves (PTV)		PBS, 2–3 coplanar or noncoplanar	50.4–54, 28–30 FX (PTV)	From DoseOpt→TEOpt with 2 or 3 beams (voxels >5% of prescription only): **CTV:** D${}_{\mathrm{mean}}$, D${}_{2\%}$ and LET${}_{\mathrm{mean}}$ equivalent. LET${}_{2\%}$ increased by 0.7 keV/μm for 3 beam TEOpt and differs <0.2 keV/μm for 2 beam TEOpt. **Brain-CTV:** D${}_{\mathrm{mean}}$, LET${}_{\mathrm{mean}}$ unchanged for 2 and 3 beams. LET${}_{2\%}$ increased by 0.8 and decreased by 2.5 GyRBE for 2 and 3 beam plans, respectively. **OARs:** LET${}_{\mathrm{mean}}$, LET${}_{2\%}$ and D${}_{2\%}$ decreased in all cases but decrease was larger for 3 beam plans. D${}_{\mathrm{mean}}$ decreased more for 3 beam plans in 2/3 cases.
Henjum et al. [63] (1-step)	-	-	-		-	-	See Table 2
Li et al. [65] (2-step)	-	-	-		-	-	See Table 3
Unkelbach et al. [30] (2-step)	5 (1,2,2)	Brain (Meningioma, Ependymoma, Base of Skull Chordoma)	Brainstem, Chiasm, Pituitary Gland		PBS, 2–6 coplanar (1 beam noncoplanar for 1 patient) ${}^{†}$, spot size 2.2–5.6 mm	50, NR (PTV)	**PTV:** All dose, BD${}_{\mathrm{mean}}$±LETOpt unchanged. **Brainstem:** BD${}_{\max}$, BD${}_{0.1cc}$ and BD${}_{0.5cc}$ decreased by 25–50% for all patients.
Hahn et al. [60] (2-step)	10	Intracranial	Brainstem, Chiasm, Optic Nerve		PBS, 2–3 coplanar and noncoplanar	54, 30 FX (Primary CTV)	For TEOpt, LETOpt, DDOpt ${}^{§}$ vs. DoseOpt - **CTV:** For all plans, equivalent dose, LET${}_{95\%}$ and LET${}_{50\%}$. D${}_{\mathrm{mean}}$ to normal brain tissue increased by <3%. **OARs:** LET${}_{\mathrm{mean}}$, LET${}_{1\%}$ decreased significantly (*p* < 0.05, <1 keV/μm) in brainstem and chiasm. Insignificant change (*p* > 0.05) in D${}_{1\%}$ and D${}_{\mathrm{mean}}$.
Bai et al. [33] and Bai et al. [31] (Ch. 4) (RO)	1	Prostate	Rectum, Bladder ${}^{†}$		PBS, 2 directly opposed coplanar beams	54, 30 FX (CTV)	From PTV-based→Dose-based RO→LETRO - **CTV:** D${}_{98\%}$, D${}_{2\%}$, BD${}_{98\%}$, BD${}_{2\%}$ robustness increases from PTV-based→RO but only BD${}_{98\%}$, BD${}_{2\%}$ robustness increased with LETRO. **Rectum:** D${}_{2\%}$, BD${}_{2\%}$, BD${}_{98\%}$ robustness increased from PTV-based→RO but unchanged with LETRO. **Bladder:** Same trend as rectum.
	1	Brain	Brainstem ${}^{†}$		PBS, 3 coplanar or noncoplanar	78, 39 FX (CTV)	**CTV:** Same trend as prostate patient. **Brainstem:** Similar robustness in D${}_{\mathrm{mean}}$ from PTV-based→RO→LETRO. Robustness of D${}_{2\%}$, BD${}_{\mathrm{mean}}$, BD${}_{2\%}$ increases from PTV-based→RO but unchanged with LETRO.
	1	H&N	NR		PBS, 3 coplanar or noncoplanar	66, 33 FX (CTV)	**CTV:** Same trend as prostate and brain patient. **OARs:** Similar trend as brainstem in brain patient for larynx and the parotid glands.
Liu et al. [80](RO)	14	H&N	Brainstem ${}^{†}$		PBS, 3 coplanar or noncoplanar	41.4–70, 20–35 FX (NR)	**CTV:** LET${}_{\mathrm{mean}}$ increased by <0.5 keV/μm, dose coverage improved up to 1.8% of the prescription dose with LETRO (where dose homogeneity = D${}_{2\%}$ − D${}_{98\%}$) **Brainstem:** D${}_{\max}$ decreased for 8 and increased for 6 patients by 5% of prescribed dose. LET${}_{\max}$ decreased 0–7 keV/μm.
Feng et al. [35] (RO)	10	Base of Skull	Temporal Lobes, necrosis occurred just outside CTV		PBS, 2–6 coplanar or noncoplanar	60–70, 30–35 FX (NR)	**CTV + 3cm Margin:** Insignificant change in D${}_{\mathrm{mean}}$, D${}_{1cc}$, D${}_{0.5cc}$, V${}_{D,50\mathrm{GyRBE}}$, BD${}_{\mathrm{mean}}$ and BD${}_{0.5cc}$ (*p* = 0.508, 0.285, 0.241, 0.314, 0.445, 0.056). Significant decrease in BD${}_{1cc}$, BD${}_{2cc}$, V${}_{\mathrm{BD},50\mathrm{Gy}}$, V${}_{\mathrm{BD},60\mathrm{Gy}}$ and V${}_{D,60\mathrm{GyRBE}}$ (*p* = 0.017, 0.022, 0.009, 0.025, 0.021).

^†^ Information extracted from figures and not reported in text. ^§^ The dirty dose method optimises dose voxels with LET > 2.5 keV/μm and LETOpt method optimises LET voxels with dose >40 GyRBE.

#### 3.3.6. LET-Based Robust Optimisation

Robust optimisation (RO) is common in PBT clinics, producing plans that are robust against proton range and patient setup uncertainties, both of which are major contributors to PBT plan uncertainty. LET-based analyses of RO plans have shown that LET can vary in OARs significantly between scenarios and varies to a lesser extent in the target [83,84]. Robustness also worsens with lower beam multiplicities [66] and use of MFO compared to SFO techniques [47], despite improving tissue sparing and conformality in PBT. Many methods of LET optimisation see an elevation of LET hotspots in the target volume, especially when defined as an objective, and shift high LET regions outside of OARs, often into normal tissue, thus inherently degrading plan robustness. As such, LET-guided RO (LETRO) has been proposed to optimise LET, whilst maintaining plan robustness [33,35,80].

Liu et al. [80] proposed a method where voxels with LET${}_{\min}$ (Appendix B) below and LET${}_{\max}$ above a user-defined constraint in the target and OAR, respectively, are penalised in addition to conventional dose objectives. The study analysed 14 patients (the biggest cohort across all LET optimisation studies included in this review), where some saw minor dosimetric changes in the target and OAR, but with no overall consensus. In the target most patients experienced a 10% increase in LET in the target, which is not dissimilar to methods proposed in other studies [17,31,77]. However, LET sparing in the OARs was far less than other optimisation methods, where improvements were generally less than 10%. Some patients experienced elevated LET in OARs; however, it occurred in a much smaller portion of the cohort. Other studies observed similar inconsistent results compared to the stepped methods discussed above [35]; however, they may be hindered by small sample sizes and high patient dependence on the efficacy of the optimisation.

The lack of improvement between conventional RO and LETRO methods may arise from a number of competing objectives in Equation (Equation 2) that cancel each other out, thus preventing the optimiser from converging to an improved solution. By nature, RO often avoids high dose gradients and distributes dose more homogeneously within a structure. Alternatively, LET optimisation will generally meet its objectives by shifting LET or BD hotspots into the target or non-OAR normal tissue, but still close to the structure boundary. Competing objectives in optimisation problems are often mitigated by the choice of penalty weights, which have been implemented in the studies discussed here [35,80].

In a comparison between conventional PTV-based, RO and LETRO plans by Bai et al. [33], a number of cases saw BD${}_{2\%}$ hotspots decrease in the target between the PTV-based and RO plan, with further reductions for LETRO. BD${}_{\mathrm{mean}}$ did not change between the three methods, while in the OARs BD${}_{\mathrm{mean}}$ decreased between the PTV-based and RO plans, but did not change with LETRO. All three plans saw a reduction of BD${}_{2\%}$. However, one major objective of this study compared to those discussed above, is that robustness of LET between scenarios was more of an objective than explicitly maximising or minimising LET.

Overall, RO would be simple to implement into existing treatment planning systems that already have RO capabilities. However, further research is needed to understand how competing objectives can be avoided. A sensitivity analysis of penalty weights between the two processes may provide insight into the correct balance to mitigate this effect.

### 3.4. Clinical Deployment of LET Optimisation

This review highlights a number of delivery-technique-based and inverse planning (objective-function-based) LET optimisation approaches, each with their own advantages and drawbacks. Many methods meet their defined objectives to varying degrees of success, mostly dependent on patient characteristics such as proximity of the target to OARs and its shape and size [64,65]. As most inverse planning methods achieve their objectives to a similar degree (regardless of being 1- or 2-step), one can consider the compatibility of each method with current clinical practice. Here, we discuss key ideas in order to guide future implementations of LET optimisation:Patients with critical structures immediately proximal to the clinical target are ideal candidates and will see the most LET reduction in OARs.Plans that often involve asymmetric beam arrangements, such as intracranial or H&N patients, should be considered at-risk due to a higher presence of LET hotspots, and thus are good candidates for LET optimisation. Whereas plans that involve directly opposed fields, such as prostate patients, should be considered lower risk. Further, modalities that do not involve LET hotspots, such as proton FLASH radiotherapy using transmission proton beams, need not be considered for LET optimisation.Clinical data are needed to properly quantify OAR-specific LET-constraints to enable LET-volume plan evaluations in conjunction with dose.Clinically achievable LET distributions are dependent on patient anatomy or the nature of the clinical target, which should also be considered in patient selection. For instance, a non-uniform or complex-shaped tumour may benefit from additional beams to increase the degrees of freedom, e.g., SPArcT.

In regard to whether there is a superior method:LET optimisation via delivery technique approaches are conceptually simpler than inverse planning and does not involve long computation times. Further systematic study of the effect that different treatment techniques have on the LET distribution will provide further insight into using this approach to its full extent.Using additional beams leads to a reduction in the overall LET distribution; however, this is at the expense of a higher volume of normal tissue receiving a low-dose bath. A more comprehensive understanding of the effects of low dose exposure to normal tissue compared to high LET will prove useful.Inverse planning approaches require long computation times but ensure a mathematically optimal plan, compared to optimising via the delivery technique. Thus, clinics would need adequate computing facilities to regularly perform LET optimisations.A 1-step inverse planning LET optimisation will give a more optimal plan than the 2-step method as the trade-off between dose and LET is managed simultaneously.The 2-step method will allow patients to be selected for LET optimisation based on their dose-optimised plan, once patient-selection protocols have been established, thus avoiding the need for computation of the LET distribution for every patient.If LET optimisation is implemented in RO, steps should be taken to ensure the objectives do not directly complete through careful selection of objective priority weights.Beam angle selection should be made with consideration given to LET. As including beam angle as an optimisable parameter in Equation (Equation 2) would increase the computation time required, a manual beam selection will still yield some improvement.Using BD or dose thresholds in place of LET alone can manage the dose–LET trade-off, ensuring that computation time is not wasted on clinically irrelevant voxels.

### 3.5. Limitations and Bias

#### 3.5.1. In the Published Data

Several recurring limitations were identified in the studies discussed throughout this review. The high computational demand of LET optimisation naturally limits the sample size on which the presented method can be tested. The largest sample size of the studies in Table 2, Table 3, Table 4, Table 5 is 14 patients [80], for which high inter-patient heterogeneities were present. Although many studies attempted to use similar target size, shape and location, significant variances still existed between studies. This limits the statistical analysis that can be performed, leading to uninformative or unreliable conclusions being drawn. Since it is often impractical to test these methods on large patient cohorts, the most effective way to avoid obfuscating the results is to consider how well the method achieves the objectives for each individual patient and identify factors that could explain any inconsistencies. Unclear presentation of results led to the exclusion of one study from this review during the data extraction stage.

Additionally, inconsistent reporting of plan parameters and evaluation made a full meta-analysis of the data infeasible. Some studies did not report parameters relating to the treatment plan or the patient, thus hindering our ability to understand if LET optimisation may or may not lead to beneficial outcomes. The majority of studies used BD or LET to extend the traditional dose-based optimisation; however, metrics were, at times, normalised in such a way that limited data interpretation and inter-study comparisons. Consistent quantitative reporting of OAR and target objectives would allow the results to be interpreted with more context. For instance, a BD decrease of 2 Gy in an OAR may be more significant if the constraint was only just met in the reference plan.

A potential source of bias is that nearly all studies are performed for patients with an OAR abutting or overlapping with CTV or PTV. These cases are considered high risk for elevated LET in OARs and will subsequently see the highest LET reductions; however, these cases are not representative of the majority population. With respect to future patient selection protocol, investigating the effect of LET optimisation across a more “generalised” cohort would improve insight into the patients that would see the most benefit.

#### 3.5.2. This Review

Several measures were taken in the formulation of this review to minimise bias; however, various limitations and sources of bias may be present. The search strategy was performed on multiple databases (MEDLINE${}^{®}$ and Scopus, as well as Google Scholar for any grey literature), with the removal of papers published in a non-English language and conference abstracts during the full-text review. The search strategy was designed with the aim to encompass the numerous terms used to describe LET optimisation, which resulted in a comprehensive list of keywords (Section A.1). Although the search strategy was reviewed by an academic librarian, it is possible that these factors lead to some articles being overlooked.

Due to the lack of consistency of the quantitative results in the published studies, a full meta-analysis was not possible. As such, quantitative improvements between the reference and LET-optimised plans were individually assessed by the authors. As this leaves the results of the review susceptible to bias, no definitive conclusion was made on which method is, quantitatively, the most effective. Instead, a judgement was made on whether the method satisfied the objectives stated in the study, e.g., *did LET${}_{\mathrm{mean}}$ increase in the target volume across all or most patients?*

## 4. Conclusions

The current status of LET optimisation in proton beam therapy is investigative; however, initial results are promising. New methods are being proposed to improve its accuracy and computation demand, although few centres currently implement it. Our current understanding of the various benefits and drawbacks suggests the following:Clinical studies with earlier and more frequent follow-ups may prove more informative than retrospective studies because the correlation between an adverse effect and LET diminishes as the endpoint progresses to a later stage. This will provide (1) a better understanding of which patient cohorts would benefit from LET optimisation and those who will not to inform future patient selection protocol and (2) the ability to quantify LET-based constraints for future implementation.Two major approaches to LET optimisation can be performed—via the delivery technique or a LET-based extension of the conventional dose objective function.Beam angles and multiplicity of a treatment plan have a strong influence on the LET distribution, whilst target size, shape and location can strongly influence the achievable LET range within. This knowledge can also be used to advise patient selection for LET optimisation.Dosimetrically, there is no clear stand-out between the methods proposed so far, but in most cases, the objectives of each study were met with varying success.Diversity between studies with respect to optimisation objectives, patient/case characteristics and reported metrics makes meta-analysis unfeasible with the current state of the literature. This would require more consistent future data reporting and would further validate or invalidate the key findings of this review.

This results thus far are highly promising and show that LET optimisation will have a role in patient treatments as the methods become more established.

## Figures and Tables

**Figure 1 cancers-15-04268-f001:**
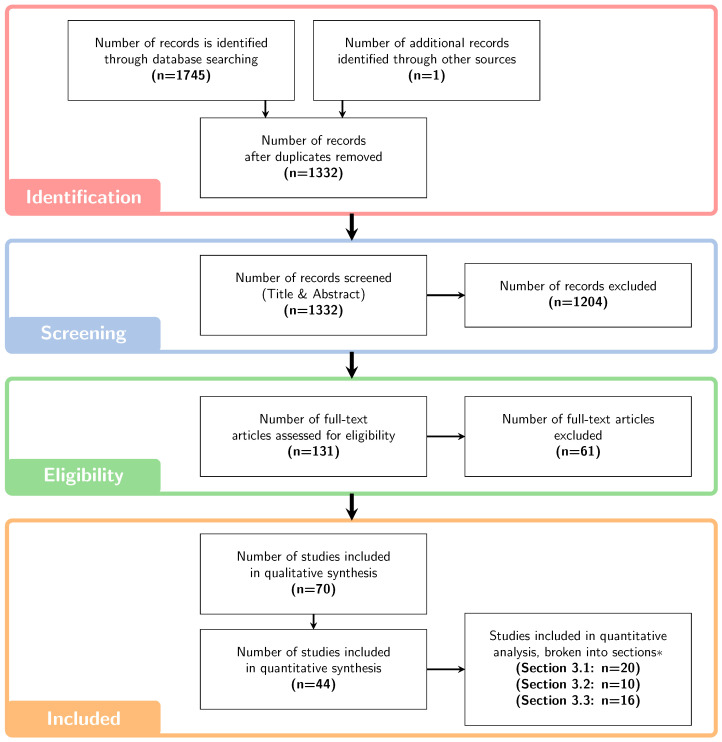
A flowchart of article collection in line with the Preferred Reporting Items for Systematic Review (PRISMA) guidelines [24]. * Two articles fall into multiple sections.

**Figure 2 cancers-15-04268-f002:**
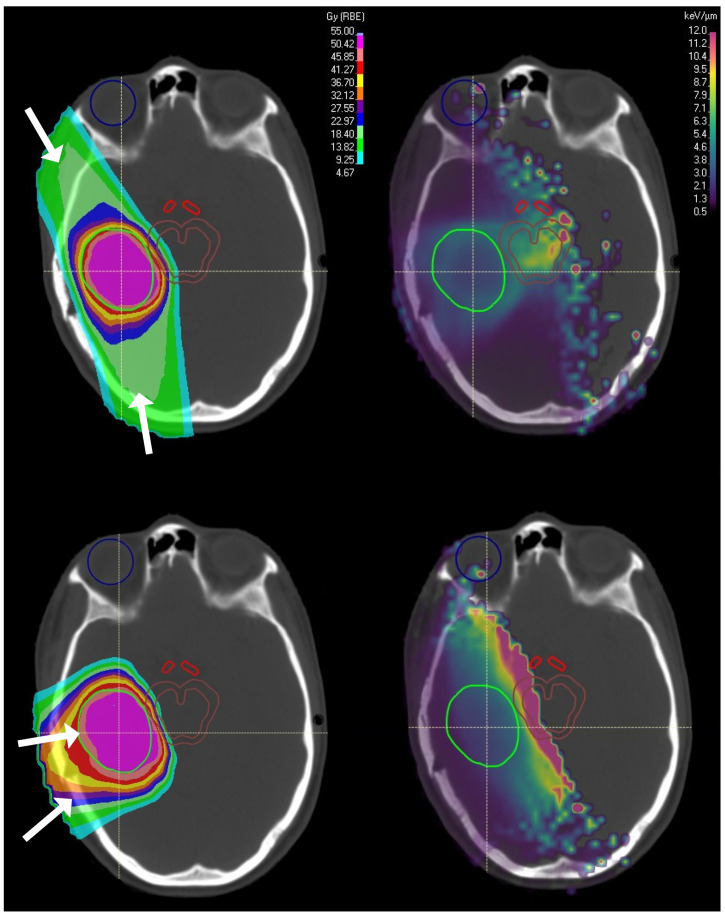
The dose (**left**) and LET (**right**) distributions for a large (**top**) and small (**bottom**) beam overlap angle. The regions of interest are the CTV (green), brainstem (brown), chiasm (red) and eye (blue). Beam angles are indicated by the white arrows. This figure was produced in Raystation v.12A (RaySearch Laboratories AB, Stockholm, Sweden).

**Figure 3 cancers-15-04268-f003:**
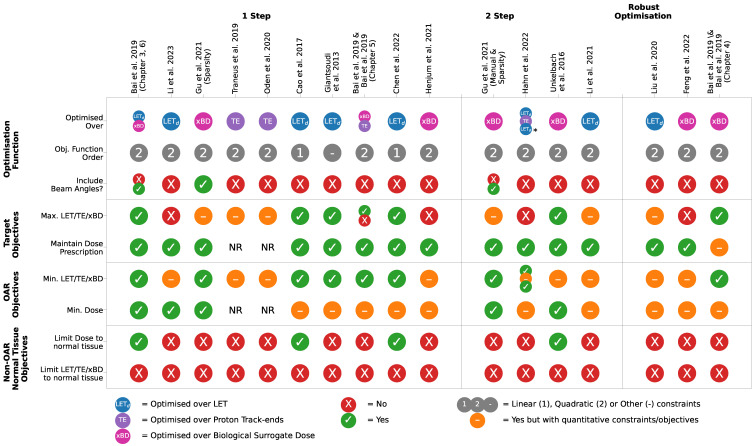
A summary of the objectives included in each study presented in Table 5. Each study is listed along the top of the figure and sorted by optimisation method [17,20,30,31,32,33,34,35,60,62,63,65,77,78,79,80]. The regions of interest are listed down the side with individual optimisation objectives. OAR, organ at risk; LET, linear energy transfer; TE, track-end; xBD, biological surrogate dose; NR, not reported. * = constraint applied.

## Data Availability

All data used to formulate this manuscript is available at https://doi.org/10.5281/zenodo.8181020, the University of South Australia database and available upon request.

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
