# Peer review of "A Systematic Review of LET-Guided Treatment Plan Optimisation in Proton Therapy: Identifying the Current State and Future Needs"

_cancers, 2023, doi:10.3390/cancers15174268_

Round 1

Reviewer 1 Report

The authors provide a systematic review on LET optimisation in proton therapy for reasons that are well explained in the paper. An extensive data search utilising several databases and applying the PRISMA guidelines have identified 70 relevant studies which are analysed. This remarkable paper is of interest for the PT community and in particular for the radiation oncologists that are not always fully aware on the issue of ‘biological’ dose as it is not modelled in their treatment plan. The authors must be commented for their notable piece of work. I have only minor concerns, namely:

Specific comments

1.      Simple summary, page 1: the effectiveness of proton treatment plans can be compromised by actually 4 significant variables

2.      Introduction section, page 1, line 28-29: PT can also be used to escalate dose delivery and increase tumor local control. Please modify the wording

3.      Introduction section, page 2, line 36- ‘massive’ would not be the first adjective characterising protons. Please modify the sentence.

4.      Introduction section, page 2, line 38-39: uncertainties in patient-setup and organ motion and its consequential effect on dose delivery corruption is not specific to proton therapy. Proton range and radiobiological effects is. It would be in the interest of the readers that this is clearly stated in this section. Change of wording necessary.

5.      Introduction section, page 2, line 48-49: The authors should explain why this universal RBE of 1.1 has been chosen and applied for dose reporting. Please modify the wording

6.      Results section, page 4, line 133: clinical benefit/where it is necessary: ill wording, please modify the sentence

Reviewer 2 Report

The article is well-organized and presents a detailed and thorough analysis of the reasons for considering LET-optimization methods in clinical practice for cancer treatment planning, despite the overall inconsistency of some results. Below are some minor comments which the authors should consider in revising their manuscript:

1. The authors have provided extensive information regarding the adverse effects of radiotherapy on healthy tissues, which are short-term. Despite the difficulty in predicting and detecting them, there is also concern about the development of secondary cancers after radiotherapy. It may be worth considering a comment regarding LET-optimization with respect to those effects as well (since LET is also used for risk assessement via the quality factor; see ICRP 60).

2. What method would the authors recommend for LET-optimization for a tumour that has difficult/complex geometry?

3. The current review does not consider the different methods for LET distribution calculations. Nevertheless, enhancing the accuracy of LET calculations would be beneficial for treatment planning and therefore, for LET-optimization. Given that LET is already an average quantity, it may deviate from more pertinent microdosimetric quantities, like lineal energy (). How would the authors hierarchy the need for LET-optimization through improved LET distribution calculations?

4. Do you think LET-optimization methods may be beneficial for FLASH radiotherapy, given that LET may play a role in the oxygen effect ?

5. The word ‘whilst’ is written too many times throughout the paper. Maybe some of them could be replaced by other words/phrases (e.g., although, on the other hand, however, etc)

Reviewer 3 Report

General comments:

The manuscript is nicely organized, and it provides an excellent compilation and summary of the current state of knowledge on a brand-new subject. It is hard to speak of scientific innovation in a review paper, but the work will be helpful because it thoroughly explains the benefits and drawbacks of the new optimization techniques. It will have the potential to open the door to agreement and give rise to fresh rules. Here follows a list of minor comments that should be addressed.

Specific comments:

Page 1

- L14: RBE-optimisations are currently unsuitable > Statement is too strong. Use "might be" instead.

- L23: protocol > protocol(s)

- L23: there is clinical evidence > Statement is too strong. See the outcomes by Niemierko, et al. "Brain necrosis in adult patients after proton therapy: is there evidence for dependency on linear energy transfer?." International Journal of Radiation Oncology* Biology* Physics 109.1 (2021): 109-119. They claim there is no clear evidence.

Page 2

- L35-36: Not clear. Rephrase or erase.

- L38: techniques > technique

- L38: which has become clinical commonplace > Is this referred to IMPT or PBS? Please rephrase.

- L46-47: this is complicated further by the use of the spread-out Bragg Peak > Please clarify this statement. Why should it be different from the previous claim about the pristine BP?

- L58: and are thus not suitable > might be (?)

Page 8

- L231: smaller beam overlap angles > If AEs have been observed where beam track-ends concentrate more, I would expect reading about "bigger beam overlap angles".

- L239: as the beam angle approaches 0 > Is this the angle in between two beams? Please clarify.

Page 9

- L265-266: at the expense of > I would also mention the lower degree of each beam-related dose uniformity. Increasing the number of beams is often related to an increased degree of modulation. Please comment on that.

Page 10

- Figure 2: large (top) and small (bottom) angle beam arrangement. > I would refer to the beam overlap angle, which would be small on top and large at the bottom.

Page 11

- Table 3: (forward-planning approaches). > This must be mentioned prior presenting the table. The planning approach could strongly affect the outcomes. Moreover the forward technique is a bit out of date nowadays. Are these findings still meaningful? Please comment on that. Moreover, the specification "LETOpt" seems to contrast with the declared optimization technique.

Page 14

- L354: The LET-optimisation methods discussed thus far use a forward-planning approach > Please refer to the comment for the header of Table 3.

Page 17

-L492-493: historically, angles are chosen such that the beam stops proximally to critical structures > This is a statement that I disagree with. Beams are chosen to leave (serial) OARs at least alongside the beam path due to the known but unaccounted for LET impact. It is less reliable to stop proximally to an OAR, particularly when range uncertainties are present.

- L501: Incomplete sentence?

Page 24

- subsection 3.3.7: I would define a separate subsection within the main Results and Discussion section (same level of Limitations and Bias subsection), since it plays as a "take home message". Well written.
